# Prevalence and factors associated with undernutrition among 15–49-year-old women in Sierra Leone: A secondary data analysis of Sierra Leone Demographic Health Survey of 2019

Nelson Onira Alema[1], Eric Nzirakaindi Ikoona[2], Mame Awa Toure[2], Oliver Eleeza[2], Amon Njenga[2], John Bosco Matovu[3], Lucy Namulemo[4,5,6], Ronald Kaluya[6], Kassim Kamara [7], Freddy Wathum Drinkwater Oyat[8], Emmanuel Olal[8,9], Judith Aloyo [8,10], David Lagoro Kitara [8,11,12]*

1 Department of Anatomy, Faculty of Medicine, Gulu University, Gulu City, Uganda, 2 ICAP at Columbia University, Freetown, Sierra Leone, 3 ICAP at Columbia University, Nairobi, Kenya, 4 Foothills Community Based Interventions, Monticello, Kentucky, United States of America, 5 Lindsey Wilson College, School of Professional Counseling, Columbia, Kentucky, United States of America, 6 Uganda Counseling and Support Services, Kampala, Uganda, 7 Directorate of Health Security and Emergencies, Ministry of Health and Sanitation, Freetown, Sierra Leone, 8 Uganda Medical Association (UMA), UMA-Acholi Branch, Gulu City, Uganda, 9 Yotkom Medical Centre, Kitgum, Uganda, 10 Rhites-N, Acholi, Gulu City, Uganda, 11 Gulu Centre for Advanced Medical Diagnostics, Research, Trainings, and Innovations (GRUDI BIONTECH INITIATIVE), Gulu City, Uganda, 12 Department of Surgery, Faculty of Medicine, Gulu University, Gulu City, Uganda

* klagoro2@gmail.com

## Abstract

### Background

Undernutrition of women of childbearing age is pertinent for maternal and offspring health. This study aimed to determine the prevalence and factors associated with undernutrition (underweight and stunting) among women of reproductive age (15–49 years) in Sierra Leone using a secondary data analysis of the 2019 Demographic Health Survey.

### Methods

Anthropometric measurements and maternal characteristics were obtained from the Sierra Leone Demographic Health Survey (SLDHS) of 2019. The heights and weights of women were measured, and BMI in kg/m² was calculated. Based on the World Health Organization's recommendations, stunting was defined as heights <145cm and being underweight as BMI <18.5kg/m². Multivariate logistic regression analyses were conducted to identify factors associated with undernutrition, with a significant level set at p<0.05.

### Results

A total of 7,514 women of reproductive age, 15–49 years were analyzed in this study. The prevalence of stunting and underweight were 1.5% (113/7514) and 6.7%(502/7,514),

Leone_standard-DHS_2019.cfm?flag=0 and within the paper and Supporting Information files.

**Funding:** The authors received no specific funding for this work.

**Competing interests:** The authors have declared that no competing interests exist.

respectively. Women with primary education had a 47% lower likelihood of being stunted (adjusted Odds Ratio [aOR] = 0.53, 95% Confidence Interval [CI]:0.30–0.94;p = 0.029) than secondary education. Women in the poorest wealth index had a 51% lower likelihood of being stunted (aOR = 0.49,95%CI:0.27–0.88;p = 0.017) than the middle wealth index. Underweight was 1.48 times more likely among women with a parity of one-to-four (aOR = 1.48,95% CI:1.08–2.03;p = 0.015) than women who never gave birth. Also, underweight was 1.41 times more likely among women who listened to radios (aOR = 1.41,95% CI:1.14–1.74;p = 0.002) than those who did not. Age groups of 15–19 years and 40–49 years had a 54% (aOR = 0.46,95%CI:0.34–0.62;p<0.001) and 34% (aOR = 0.66,95%CI:0.45–0.97;p = 0.035) lower likelihood of being underweight than 20-29-year age group, respectively. Women with primary education had a 26% lower likelihood of being underweight (aOR = 0.74,95%CI:0.56–0.99;p = 0.042) than those with secondary education. However, none of the wealth indices was significantly associated with being underweight.

## Conclusion

The prevalence of underweight and stunting among women of reproductive age (15–49 years) in Sierra Leone was lower than regional and world data. This study highlights similarities and differences in this population's prevalence and factors associated with undernutrition. Underweight and stunting were less likely in women with primary education, while parity of one to four and listening to radios were significantly associated with being underweight. Further trend studies using DHS data from 2010, 2014, and 2019 are warranted to understand the dynamics of undernutrition among women (15–49 years) in Sierra Leone.

## Introduction

Undernutrition, characterized by deficiencies in calories, proteins, vitamins, minerals, poor health, and social conditions, poses a significant health challenge for millions of women and adolescent girls worldwide [1]. Adequate nutrition is crucial for women's overall health and has far-reaching implications for the well-being of their children [1]. Children born to malnourished women are at higher risk of cognitive impairments, stunted growth, increased susceptibility to infections, and elevated morbidity and mortality rates throughout their lives [1].

Undernutrition remains a pressing global health issue, encompassing being underweight, wasted, stunted, and with deficiencies in essential minerals and vitamins [2]. Research indicates that women with a body mass index (BMI) below $18.5 \text{kg/m}^2$ in developing countries face an escalating mortality risk and heightened vulnerability to illnesses [2–4]. Accordingly, the impact of undernutrition extends beyond women's health, affecting the well-being of their children [5, 6]. This scenario perpetuates a cycle of undernutrition that spans generations [5, 6], especially in countries like Sierra Leone, where social and biological factors such as civil unrest, poverty, epidemic outbreaks, and food insecurity contribute to women's vulnerability to undernutrition [5]. In addition, numerous individual, household, and community factors influence women's nutritional health status [5, 6].

On the one side, stunting is a consequence of complex interactions among household, environmental, socioeconomic, and cultural factors [7–9]. It has detrimental effects such as susceptibility to infections, impaired cognitive and motor development, and elevated risks of developing non-communicable diseases (NCDs) later in life [7–9]. Also, research have shown

that individuals who experience stunting during childhood were more likely to face challenges such as poor cognitive function, lower educational performance, reduced adult wages, decreased productivity, and increased risks of nutrition-related chronic diseases in adulthood [10]. Therefore, ensuring adequate nutrition for a person is a fundamental foundation for individual and population health [11–16]. Furthermore, maternal undernutrition, underweight, and stunting have been linked to adverse maternal health conditions, such as chronic energy deficiency, cesarean delivery, pre-eclampsia, anemia, decreased productivity, mental health issues, and adverse pregnancy outcomes [11–16]. On the other end of the malnutrition spectrum, overweight and obesity pose significant health risks for women, including a higher likelihood of developing hypertension, diabetes, cardiovascular diseases, and stroke [17–20].

The determinants of undernutrition in women encompass many factors, including community-level water, sanitation, and hygiene (WASH) practices [21, 22], food stability status [23], household income and wealth, women's level of education, age at first marriage, age at first delivery, multiparity, short birth intervals, and land ownership [19, 24–28]. Therefore, identifying maternal malnutritional prevalence levels and determinants is crucial for targeted interventions and resource allocation in resource-limited settings [19, 21–28].

Despite the significance of understanding maternal nutritional status, limited research have been conducted in Sierra Leone because, very often the focus is solely on malnutrition determinants in children and young adolescents. The present study addresses this research gap by investigating the factors associated with undernutrition among women of reproductive age (15–49 years) in Sierra Leone by utilizing data from the Sierra Leone Demographic Health Survey (SLDHS-2019). Findings of this study hold essential policy implications from a global health perspective and specifically for Sierra Leone, and helps in monitoring progress towards sustainable development goals (SDGs) and regional nutritional strategies. Moreover, the study can guide the allocation of limited resources by the Government and health stakeholders to improve the nutritional and health status of women and infants in Sierra Leone.

In using data from a population-based cohort of women of childbearing age in Sierra Leone, this study aimed to determine the prevalence and factors associated with undernutrition (underweight and stunting) among women of reproductive age (15–49 years) based on the 2019 Demographic Health Survey.

## Methods

### Study design

The SLDHS-2019 was conducted as a countrywide representative cross-sectional survey led by the Bureau of Statistics of Sierra Leone (Stats SL) with technical assistance from ICF through DHS programs [29]. This survey was funded by the United States Agency for International Development (USAID) [29].

### Study sites

This study was conducted in all four provinces and western areas of Sierra Leone [29].

### Sampling and study population

The sampling of the study respondents was based on the 2015 Population and Housing Census of the Republic of Sierra Leone [30]. The 2015 Population and Housing Census provided the ready-made sampling frame for the SLDHS-2019 [30]. Sierra Leone is administratively divided into four provinces and western areas (urban and rural), sixteen districts, and 190 chiefdoms [30–32]. Each district is subdivided into chiefdoms/census wards, and each chiefdom/census

ward is subdivided into sections [30–32]. Also, the 2015 Population and Housing Census subdivided each locality into convenient census areas; the Enumeration Areas (EAs) [30, 33]. The EAs were the primary sampling units (PSUs) and clusters for the SLDHS-2019 [30–35]. The list of EAs from the 2015 census formed the basis for estimating the number of households required for the study and for classifying EAs (clusters) into urban/rural for the SLDHS-2019 sampling frame [30, 31, 34, 35]. Furthermore, the SLDHS-2019 employed a two-stage stratified sampling design, where stratification was achieved by classifying each district into urban and rural areas [34, 35].

So, thirty-one sampling strata were created, and samples were selected independently in each stratum via a two-stage selection process [34, 35]. Thus, implicit stratifications were achieved at each lower administrative level by sorting the sampling frame before sample selection according to administrative order and using a probability proportional-to-size selection during the first sampling stage [34, 35].

Thus, five hundred and seventy-eight (578) EAs were selected using a probability proportional to EA size [34, 35] in the first stage of the selection process. In addition, the enumeration area size was determined by the number of households residing in it and a household listing operation was performed in all selected enumeration areas [34, 35]. The resulting lists of households served as a sampling frame for selecting households in the second stage of the survey [34, 35].

In the second stage's selection, a fixed number of twenty-four households was chosen in every cluster through an equal probability systematic sampling, resulting in a total sample size of approximately 13,872 households distributed in the 578 clusters [34, 35]. The household listing at this stage was conducted using computer tablets, and households were randomly selected through computer programming [34, 35].

The survey interviewed only pre-selected households in the clusters, and no replacements or changes of the selected households were allowed in the implementation stage of the survey to prevent selection bias in the study population [34, 35]. Due to the non-proportional allocation of samples to the sixteen districts in Sierra Leone and the possible differences in response rates, sample weights were calculated, added to the data file, and applied so that the results would be representative at national and domain levels [34, 35]. Further, because the SLDHS-2019 sample was a two-stage stratified cluster sampling, sample weights were calculated separately at each sampling stage based on sampling probabilities [34, 35]. After that, the SLDHS-2019 included all women aged 15–49 in the sampled households [34, 35].

Permanent residents in the selected homes and visitors who stayed overnight before the survey were eligible for interviews in the household [34, 35]. The Man's Questionnaire covered the identification of respondents, background information, reproductive, contraceptive, marriage and sexual activities, fertility preferences, employment status, gender roles, HIV and AIDS, and other health-related issues [35]. The Biomarker Questionnaire covered the identification of respondents, weights, heights, and hemoglobin measurements for children aged 0–5 years, weights, heights, HIV testing, and hemoglobin measurements for women aged 15–49 years [35]. The Fieldworker Questionnaire covered background information on each field worker [35].

## Anthropometric measurements

The weight of respondents was recorded in kilograms (kg) to the nearest decimal point and was measured using an electronic scale (SECA 878) [34, 35]. Participants' heights were measured using a stadiometer in centimeters (cm) to one decimal point [34, 35]. The Body Mass Index (BMI) of respondents was calculated in kg/m$^2$ using weights (in kilograms) and heights

(meters) of women of reproductive age (15–49 years) and classified according to WHO criteria as underweight ($<18.5\text{kg/m}^2$); normal weight ($18.5–24.9\text{kg/m}^2$); overweight ($25.0–29.9\text{kg/m}^2$); obese ($\geq30.0\text{kg/m}^2$ and $\leq50.0\text{kg/m}^2$), and overnutrition ($\geq25.0\text{kg/m}^2$ and $\leq50.0\text{kg/m}^2$) [36, 37].

## Wealth Index (WI)

To calculate each household's wealth, we used the wealth index (WI) as a proxy indicator of household wealth [35]. This composite index used household key asset ownership variables to calculate each household wealth index from the SLDHS-2019 data [35]. These variables were the characteristics of the household's dwelling unit, for example, the source of water, type of toilet facilities, type of fuel used for cooking, number of rooms, ownership of livestock, possessions of durable goods, mosquito nets, and primary materials for the floor, roof, and walls of the dwelling place [35]. Each respondent's household wealth index was calculated using computer analysis of household composite factors [35]. It was then categorized into five quintiles: poorest, poorer, middle, richer, and richest wealth indices. (Table 1).

## Operational definitions

**Body Mass Index (BMI):** Weight in kilograms divided by heights in meters squared ($\text{kg/m}^2$).

 **Underweight:** BMI $<18.5\text{kg/m}^2$

 **Overweight:** BMI $\geq25.0\text{kg/m}^2$ and $\leq29.9\text{kg/m}^2$

 **Obese:** BMI $\geq30.0\text{kg/m}^2$ and $\leq50.0\text{kg/m}^2$

 **Overnutrition (Overweight and obese):** BMI $\geq25.0\text{kg/m}^2$ and $\leq50.0\text{kg/m}^2$.

 **Undernutrition (Underweight and Stunting)** where stunting is heights of participants $<145\text{cm}$ [37].

 **Enumeration Area (Clusters):** An EA is a geographic area consisting of a convenient number of dwelling units that serve as a counting unit for the survey.

## Data collection

Data collection for this survey was conducted from May 14, 2019, to August 31, 2019 [29]. The primary sampling unit (PSU), a cluster, was based on enumeration areas (EAs) obtained from the 2015 EA population census sampling frame [29]. The SLDHS-2019 used five validated questionnaires for the thematic parts of the survey [29]. The household Questionnaire collected data on the household environment, assets, and basic demographic information of household members. The Woman's Questionnaire collected data on women's reproductive health information, domestic violence, and nutrition indicators [29]. The Man's Questionnaire collected data on men's background characteristics (age, education, employment status, marital status, media exposure, and place of residence, while the Biomarker Questionnaire collected data on anthropometry and blood tests for mothers and children (0–5 years), and the Fieldworker Questionnaire collected data on the background information of fieldworkers [34, 35].

This secondary data analysis included women of reproductive age, 15–49 years, whose anthropometric characteristics were recorded with their consent. Trained health technicians were deployed to measure the heights and weights of respondents to ensure the quality of the anthropometric measurements [29].

Out of the weighted sample of 15,574 women in the dataset, 7,514 anthropometric measurements were included in the survey design, while 8,060 had invalid weight and height measurements due to erroneous and ineligible measurements. The weight and height measurements are vital for calculating the BMI of each respondent, which was finally used for assessing the nutritional status of each respondent.

**Table 1. Socio-economic and demographic characteristics of women of reproductive age (15–49 years) in Sierra Leone.**

| Variables | Frequency (n = 7,514) | Percent (%) |
|---|---|---|
| **Ages (years)** | | |
| 15–19 | 1,616 | 21.5 |
| 20–29 | 2,528 | 33.6 |
| 30–39 | 2,048 | 27.3 |
| 40–49 | 1,322 | 17.6 |
| **Parity** | | |
| Never gave birth | 1,895 | 25.2 |
| One to four | 3,892 | 51.8 |
| Five and above | 1,727 | 23.0 |
| **Type of residence** | | |
| Urban | 3,092 | 41.1 |
| Rural | 4,422 | 58.9 |
| **Sex of the head of household** | | |
| Male | 5,356 | 71.3 |
| Female | 2,158 | 28.7 |
| **Household size** | | |
| Less than six | 2,995 | 39.9 |
| Six and above | 4,519 | 60.1 |
| **Work status** | | |
| Not working | 2,280 | 30.3 |
| Working | 5,234 | 69.7 |
| **Marital status** | | |
| Married | 4,795 | 63.8 |
| Not married | 2,719 | 36.2 |
| **Regions of Sierra Leone** | | |
| East | 1,579 | 21.0 |
| North | 1,822 | 24.2 |
| Northwest | 1,026 | 13.7 |
| South | 1,831 | 24.4 |
| Western | 1,256 | 16.7 |
| **Levels of education** | | |
| No formal education | 3,571 | 47.5 |
| Primary | 1,017 | 13.5 |
| Secondary | 2,641 | 35.2 |
| Higher | 285 | 3.8 |
| **Wealth Indices** | | |
| Poorest | 1,533 | 20.4 |
| Poorer | 1,428 | 19.0 |
| Middle | 1,531 | 20.4 |
| Richer | 1,634 | 21.7 |
| Richest | 1,388 | 18.5 |
| **BMI categories (kg/m$^2$)** | | |
| Underweight ($<$18.5) | 502 | 6.7 |
| Normal weight (18.5–24.9) | 4,974 | 66.2 |
| Overweight (25.0–29.9) | 1,479 | 19.7 |
| Obese ($\geq$30.0) | 559 | 7.4 |

*(Continued)*

**Table 1.** (Continued)

| Variables | Frequency (n = 7,514) | Percent (%) |
|---|---:|---:|
| **Watching Television** | | |
| Yes | 1,889 | 25.1 |
| No | 5,625 | 74.9 |
| **Listening to radios** | | |
| Yes | 3,142 | 41.8 |
| No | 4,372 | 58.2 |
| **Reading of magazines** | | |
| Yes | 489 | 6.5 |
| No | 7.025 | 93.5 |
| **Smoking of cigarettes** | | |
| Yes | 224 | 3.0 |
| No | 7,290 | 97.0 |
| **Alcohol use** | | |
| No response recorded | 3,766 | 50.1 |
| Yes | 667 | 8.9 |
| No | 3,081 | 41.0 |

The data source is SLDHS-2019.

In Table 1 the majority of women of reproductive age (15–49 years) in Sierra Leone were in the 20-29-year age group 2528/7514(33.6%); parity of one-to-four 3892/7514(51.8%); of rural residence 4422/7514(58.9%); male-headed households 5356/7514(71.3%); the household size of six and above 4519/7514(60.1%); working class 5234/7514 (69.7%); married 4,795/7514(63.8%); from the South 1831/7514(24.4%); had no formal education 3571/7514(47.5%); richer wealth index 1634/7514(21.7%); normal weight 4974/7514(66.2%); did not watch television 5625/7514(74.9%); did not listen to radios 4372/7514(58.2%); did not read magazines 7025/7514(93.5%), did not smoke cigarettes 7290/7514(97.0%), and did not respond to the alcohol use question 3766/7514(50.1%).

In some respondents' results, heights and weights were not well recorded (n = 7548) and some respondents refused to have weights and heights taken (n = 512), and we could only obtain completed anthropometric measurements for 7,514 women who were not lactating, non-pregnant, and not post-menopausal women. In the final analysis, a weighted sample of 7,514 was used in our secondary data analysis, as summarized in Fig 1 and Table 1. A complete protocol with detailed explanations about data collection processes and sampling is available online [29].

## Outcome variables

The first outcome variable for this study was stunting among women (15–49 years). Stunting was defined as heights of <145cm ± Standard Deviations (SD) from the median value set by the World Health Organization (WHO) [36, 37]. The second outcome variable was underweight (BMI <18.5kg/m$^2$).

## Independent variables

The independent variables in this study were derived from previous studies, the WHO stunting framework, underweight, normal weight, and available information in the SLDHS-2019 database. We included sixteen independent variables (age, parity, type of residence, sex of household head, Household size, work status, marital status, regions of Sierra Leone, level of

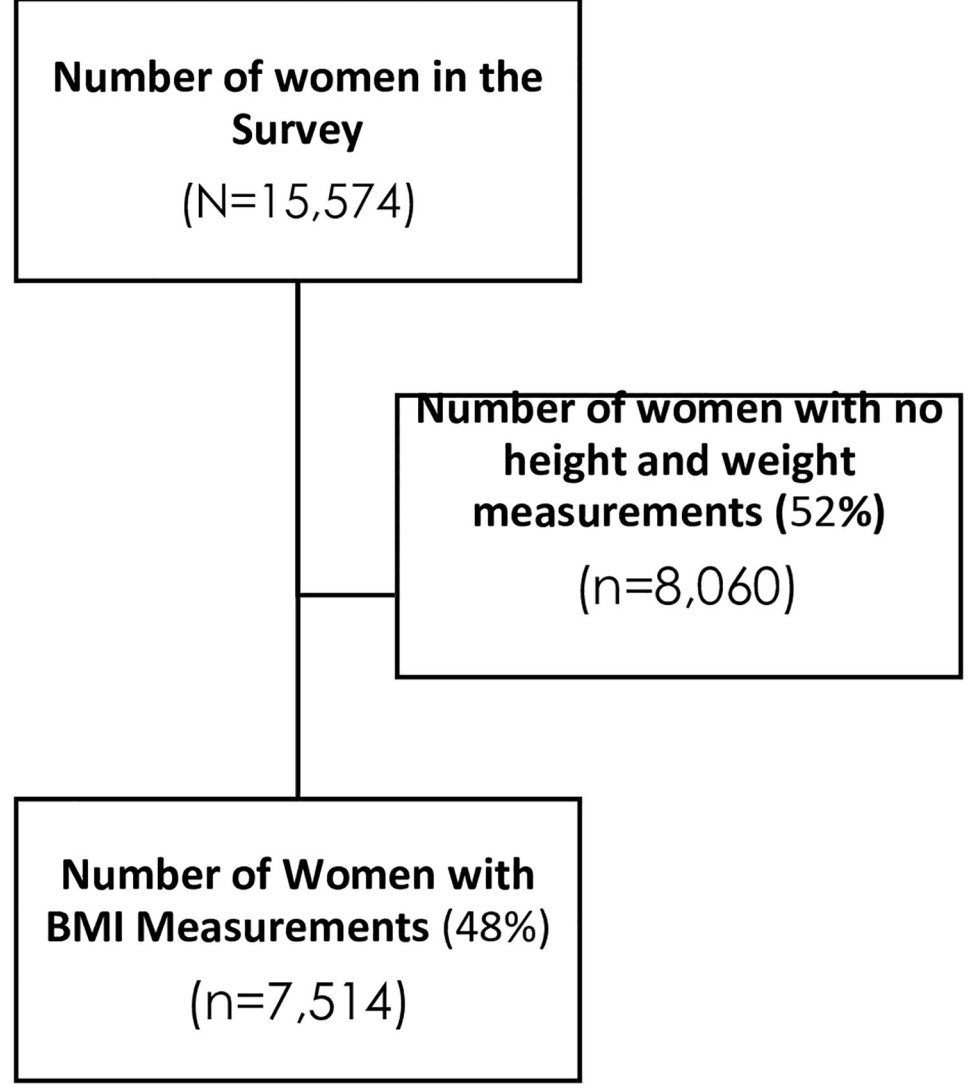

**Fig 1. Flow chart for the study.**

education, wealth index, BMI categories, listening to radios, reading of magazines, watching television, smoking cigarettes, and alcohol use) in this data analysis.

## Women's characteristics

Parity (categorized as para 0, para one-to-four, and five and above), work status (categorized as working-class versus not working), marital status (categorized as married versus not married/single), levels of education (categorized as no education, primary, secondary, and higher), age groups (categorized as 15–19, 20–29, 30–39, and 40–49 years), woman's stunting status (defined as heights <145cm for stunted and ≥145cm for not stunted women), and woman's BMI classification as normal BMI ($18.5–24.9kg/m^2$) and underweight ($<18.5kg/m^2$) [36, 37].

### Household characteristics

These characteristics include; regions of Sierra Leone (Northwest, Eastern, Western, Southern, and Northern); household wealth indices (categorized as richest, richer, middle, poorer, and poorest); sex of the head of household (female versus male); household size (less than six versus six and above); residency (urban versus rural); watching television (yes versus no); reading of magazines (yes versus no); listening to radios (yes versus no); smoking cigarettes (yes versus no); and alcohol use (yes versus no).

**Ethical approval.** This survey protocol was approved by the Sierra Leone Ethics and Scientific Review Committee (SLESRC) and the ICF Institutional Review Board. In addition, this study was conducted according to their institutional guidelines, where written informed consent was obtained from each adult respondent. For those under 18 years, assent was obtained in the presence of their parents or legal representatives.

### Data analysis

Frequency tables and proportions/percentages were used to describe summaries of categorical variables, while means and standard deviations (±SD) were used for continuous variables. Sample weights were used to account for unequal probability sampling in different study population strata and ensured the representativeness of the survey results at all levels [29]. Statistical software SPSS version 25.0 Statistical software complex samples package incorporating all variables in the analysis plan was used to account for the multistage sampling design inherent in the DHS dataset, including individual sample weight, sample strata for sampling errors/design, and cluster numbers [38–42].

Using a complex sample package ensured the sampling design was incorporated into the analysis, leading to accurate and reliable results. Cross tabulations were conducted, and associations between socio-demographic and economic characteristics, and women's nutritional status (stunting and underweight), including their Odds ratios (OR) and P-values, were presented.

To assess associations of each independent variable with dependent variables (stunting and underweight), a bivariate logistic regression analysis was conducted, and Crude Odds Ratios (COR), at 95% Confidence Intervals (CI) and P-values were presented. Independent variables which were significant at bivariate level, and those with P-values ≤0.20 were included in the final multivariate logistic regression analysis model for each dependent variable. The final regression model excluded variables with P-values above 0.201 at bivariate level. The final multivariate logistic regression analysis calculated the adjusted Odds Ratios (aOR), at 95% Confidence Intervals (CI), and corresponding P-values with a statistical significance level set at 0.05.

**Sensitivity analyses.** The sensitivity analysis for stunting was conducted by excluding women with parity of five and above in the multivariate logistic regression model, as it had only 23(1.3%) stunted women. By excluding them from the final regression model, the other factors remained significant, and no substantial changes were observed in the strength of associations. A similar statistical approach was used for studying the sensitivity of underweight data in this study population. Cross tabulations were conducted, and associations between socio-demographic characteristics and women's nutritional status (underweight versus normal weight), including their aOR at 95% CI and P-values, were presented with no significant differences observed after excluding women with BMI ≤15.0kg/m$^2$.

## Results and discussions

This study was a secondary data analysis of the demographic health survey conducted in Sierra Leone in 2019 and included 7,514 women of reproductive age, 15–49 years in the final analysis

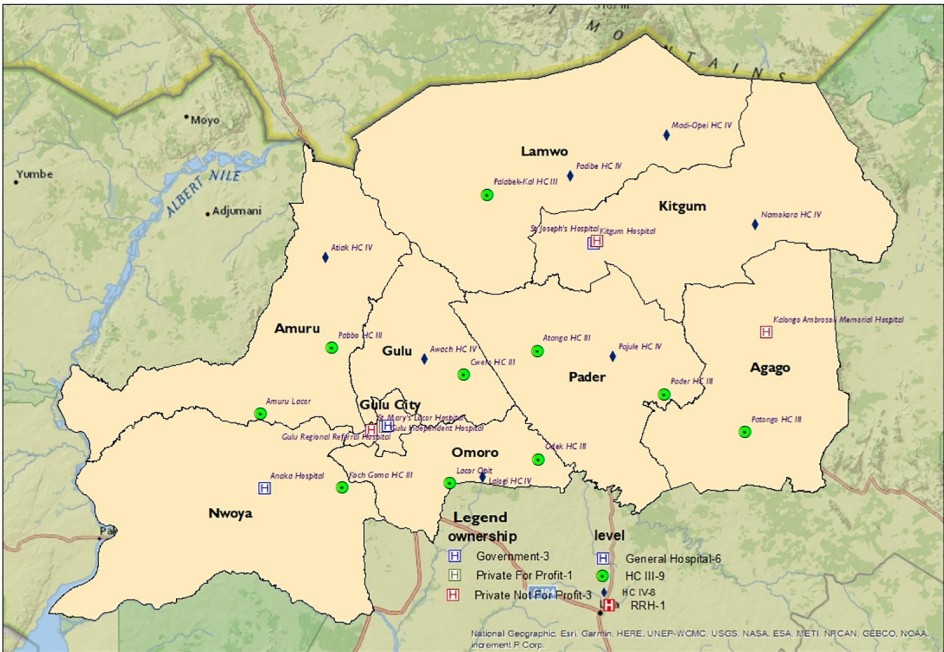

**Fig 2. The heights of women in the reproductive age (15–49 years) in Sierra Leone.** Fig 2 shows normally distributed heights among women aged (15–49) years in Sierra Leone. The mean height was 157.6 cm, SD±6.3. The data source is SLDHS of 2019.

(Table 1, Figs 1 and 2). Among the women, the majority belonged to the 20-29-year age group, accounting for 33.6%(2528/7514) of the total population. Women with parity of one to four represented just over 50%(51.8%; 3892/7514) of the total study population, while most respondents resided in rural areas, 58.9%(4,422/7,514) of Sierra Leone. Male-headed households constituted slightly over two-thirds of the study population at 71.3%(5,356/7,514). Moreover, households with a size of six or more individuals constituted the majority at 60.1%(4,519/7,514) (Table 1).

Most study population were working women, representing 69.7%(5,234/7,514) of the population. Among marital status categories, married women constituted 63.8%(4,795/7,514).

Regionally, women from the south of Sierra Leone constituted the largest proportion at 24.4%(1,831/7,514), followed by the north at 24.2%(1,822/7,514), the east at 21.0%(1,579/7,514), the west at 16.7%(1,256/7,514), and the northwest at 13.7%(1,026/7,514). At educational level, the majority had no formal education, accounting for 47.5%(3,571/7,514), followed by secondary education at 35.2%(2,641/7,514), primary education at 13.5%(1,017/7,514), and a smaller proportion with higher education at 3.8%(285/7,514) (Table 1).

Regarding wealth indices, 21.7%(1,634/7,514) of the women were in the richer wealth index category, followed by the poorest and middle at 20.4%(1,533/7,514) each, poorer wealth index at 19.0%(1,482/7,514), and the smallest proportion were among the richest wealth index at 18.5%(1,388/7,514) (Table 1).

In terms of BMI categories, most women had normal BMI, accounting for 66.2%(4,974/7,514), followed by overweight at 19.7%(1,479/7,514), obese at 7.4%(5,59/7,514) and the smallest proportion among underweight women at 6.7%(502/7,514) (Table 1).

Regarding social activities, most participants did not watch television, accounting for 74.9% (5,625/7,514). Furthermore, most participants did not listen to radios 58.2%(4,372/7,514) and

did not read magazines 93.5%(7,025/7,514). Additionally, most women did not smoke cigarettes 97.0%(7,290/7,514) and did not respond to alcohol use question 50.1%(3,766/7,514) (Table 1).

## The prevalence of stunting in the study population

Out of 15,574 women in the SLDHS-2019, 48%(7,514/15,574) had valid height measurements. The mean height was 157.6cm with a standard deviation (SD) of ±6.3cm. The minimum recorded height was 107.7cm, and the maximum was 186.2cm (Fig 2). The overall prevalence of stunting in the study population was 1.5%(113/15,574) (Table 2).

## The prevalence of underweight

Among the study population of women (n = 7,514), the mean BMI was 23.8kg/m$^2$ (SD±4.7). The prevalence of underweight was 6.7%(502/7,514), with a minimum BMI recorded at 12.8kg/m$^2$ (Fig 3). Within the underweight category, two outlier BMI values were 12.8kg/m$^2$ and 14.5kg/m$^2$, each representing 0.03% of the total study population. These outlier BMIs were situated on the left side of the normal distribution of underweight (Table 2).

## Factors associated with stunting among women of reproductive age (15–49 years) in Sierra Leone

The study showed that primary education and being in the poorest wealth index were less likely factors for being stunted among the study population. Women with primary education had a 47% lower likelihood of being stunted (aOR = 0.53,95%CI:0.30–0.94;p = 0.029) than those with secondary education. Similarly, women in the poorest wealth index had a 51% lower likelihood of being stunted (aOR = 0.49,95%CI:0.27–0.88;p = 0.017) than those in the middle wealth index. Other factors such as parity, residence (urban or rural), sex of the household head, household size, work status, marital status, regions of residence, listening to radios, reading of magazines, watching television, alcohol use, and smoking cigarettes did not significantly affect stunting among study participants (Table 2).

## Factors associated with underweight among women (15–49 years) in Sierra Leone

After adjusting for individual characteristics in the final multivariate logistic regression model, the determinants of being underweight among Sierra Leonean women (15–49 years) were: Women with parity of one-to-four had a 1.48 times more likelihood of being underweight (aOR = 1.48, 95%CI:1.08–2.03; p = 0.015) than women who never gave birth. Women who listened to radios were 1.41 times more likely to be underweight (aOR = 1.41,95%CI:1.14–1.74; p = 0.002) than those who did not. However, being in the age group of 15–19 years was associated with a 54% lower likelihood of being underweight (aOR = 0.46, 95% CI:0.34–0.62; p<0.001) than the 20-29-year age group and being in the age group of 40–49 years was associated with a 34% lower likelihood of being underweight (aOR = 0.66, 95%CI:0.45–0.97; p = 0.035). Furthermore, having primary education was associated with a 26% lower likelihood of being underweight (aOR = 0.74, 95% CI:0.56–0.99; p = 0.042) than secondary education. However, none of the wealth indices showed a significant association with being underweight in this study population (Table 3).

This population-based study provides valuable insights into the prevalence and factors associated with underweight and stunting among women of reproductive age (15–49 years) in Sierra Leone (Table 1, Figs 1–3). The prevalence of being stunted among women (15–49 years)

**Table 2. Bi-and multivariate analysis of stunting among women (15–49 years) in SLDHS-2019.**

| Variables | Stunted (n = 113) (n, %) | Not stunted (n = 7,401) (n, %) | Unadjusted COR | 95% CI | p-value | aOR | 95% CI | p-value |
|---|---|---|---|---|---|---|---|---|
| **Age groups (years)** | | | | | | | | |
| 20–29 | 33(1.3) | 2495(98.7) | Reference | | | Reference | | |
| 15–19 | 35(2.2) | 1582(97.8) | 0.597 | 0.370–0.965 | 0.035 | 0.815 | 0.437–1.520 | 0.419 |
| 30–39 | 32(1.6) | 2016(98.4) | 0.833 | 0.511–1.360 | 0.465 | 0.936 | 0.533–1.644 | 0.520 |
| 40–49 | 13(1.0) | 1309(99.0) | 1.332 | 0.699–2.539 | 0.384 | 1.559 | 0.741–3.277 | 0.818 |
| **Parity** | | | | | | | | |
| Never gave birth | 38(2.0) | 1857(98.0) | Reference | | | Reference | | |
| One to four | 52(1.3) | (3840(98.7) | 1.511 | 0.991–2.304 | 0.055 | 1.489 | 0.792–2.801 | 0.216 |
| Five and above | 23(1.3) | 1704(98.7) | 1.516 | 0.900–2.555 | 0.118 | 1.524 | 0.659–3.524 | 0.324 |
| **Residence** | | | | | | | | |
| Rural | 80(1.8) | 4342(98.2) | Reference | | | Reference | | |
| Urban | 33(1.1) | 3059(98.9) | 1.708 | 1.136–2.569 | 0.010 | 1.257 | 0.614–2.572 | 0.531 |
| **Sex of the household head** | | | | | | | | |
| Male | 76(1.4) | 5280(98.6) | Reference | | | | | |
| Female | 37(1.7) | 2121(98.3) | 0.825 | 0.555–1.226 | 0.342 | | | |
| **Household size** | | | | | | | | |
| Six and above | 59(1.3) | 4460(98.7) | Reference | | | Reference | | |
| Less than six | 54(1.8) | 2941(98.2) | 0.72 | 0.497–1.045 | 0.084 | 0.761 | 0.518–1.112 | 0.166 |
| **Work status** | | | | | | | | |
| Not working | 33(1.4) | 2247(98.6) | Reference | | | | | |
| Works | 80(1.5) | 5154(98.5) | 0.946 | 0.946–1.424 | 0.791 | | | |
| **Marital status** | | | | | | | | |
| Not married | 48(1.8) | 2671(98.2) | Reference | | | Reference | | |
| Married | 65(1.4) | 4730(98.6) | 1.308 | 0.898–1.905 | 0.162 | 1.303 | 0.763–2.224 | 0.333 |
| **Region of residence** | | | | | | | | |
| East | 24(1.5) | 1555(98.5) | Reference | | | Reference | | |
| North | 25(1.4) | 1797(98.6) | 1.109 | 0.631–1.950 | 0.718 | 1.167 | 0.656–2.074 | 0.600 |
| Northwest | 8(0.8) | 1018(99.2) | 1.964 | 0.879–4.389 | 0.100 | 1.908 | 0.846–4.302 | 0.119 |
| South | 43(2.3) | 1788(97.7) | 0.642 | 0.388–1.062 | 0.085 | 0.695 | 0.415–1.162 | 0.165 |
| Western | 13(1.0) | 1243(99.0) | 1.476 | 0.748–2.910 | 0.261 | 0.821 | 0.368–1.831 | 0.630 |
| **Level of education** | | | | | | | | |
| Secondary | 32(1.2) | 2609(98.8) | Reference | | | Reference | | |
| No formal education | 56(1.6) | 3515(98.4) | 0.770 | 0.497–1.192 | 0.241 | 0.624 | 0.351–1.107 | 0.107 |
| Primary | 25(2.5) | 992(97.5) | 0.487 | 0.287–0.825 | 0.008 | 0.531 | 0.300–0.938 | 0.029 |
| Higher | 0(0.0) | 285(100.0) | 19814180 | 0 | 0.994 | 12203543 | 0 | 0.994 |
| **Wealth Indices** | | | | | | | | |
| Middle | 17(1.1) | 1514(98.9) | Reference | | | Reference | | |
| Poorest | 41(2.7) | 1492(97.3) | 0.409 | 0.231–0.722 | 0.002 | 0.485 | 0.268–0.880 | 0.017 |
| Poorer | 21(1.5) | 1407(98.5) | 0.752 | 0.395–1.432 | 0.386 | 0.778 | 0.404–1.497 | 0.452 |
| Richer | 24(1.5) | 1610(98.5) | 0.753 | 0.403–1.408 | 0.374 | 0.654 | 0.302–1.417 | 0.282 |
| Richest | 10(0.7) | 1378(99.3) | 1.547 | 0.706–3.391 | 0.275 | 1.068 | 0.377–3.026 | 0.902 |
| **Watching television** | | | | | | | | |
| No | 96(1.7) | 5529(98.3) | Reference | | | Reference | | |
| Yes | 17(0.9) | 1872(99.1) | 1.912 | 1.139–3.211 | 0.014 | 1.385 | 0.744–2.578 | 0.304 |
| **Listens to radio** | | | | | | | | |
| No | 74(1.7) | 4298(98.3) | Reference | | | Reference | | |
| Yes | 39(1.2) | 3103(98.8) | 1.37 | 0.927–2.204 | 0.114 | 0.902 | 0.585–1.392 | 0.642 |

*(Continued)*

**Table 2.** (Continued)

| Variables | Stunted (n = 113) (n, %) | Not stunted (n = 7,401) (n, %) | Unadjusted COR | 95% CI | p-value | aOR | 95% CI | p-value |
|---|---|---|---|---|---|---|---|---|
| **Reading of magazines** | | | | | | | | |
| No | 108(1.5) | 6917(98.5) | Reference | | | | | |
| Yes | 5(1.0) | 484(99.0) | 1.511 | 0.614–3.722 | 0.369 | | | |
| **Smokes cigarettes** | | | | | | | | |
| No | 109(1.5) | 7181(98.5) | Reference | | | | | |
| Yes | 4(1.8) | 220(98.2) | 0.835 | 0.305–2.285 | 0.725 | | | |
| **Alcohol use** | | | | | | | | |
| No | 45(1.5) | 3036(98.5) | Reference | | | | | |
| Yes | 8(1.2) | 659(98.8) | 1.221 | 0.573–2.602 | 0.605 | | | |

aOR: adjusted Odds Ratio; CI: Confidence Interval; COR: Crude Odds Ratio; SLDHS: Sierra Leone Demographic and Health Survey.

In Table 2, the correlates of stunting among Sierra Leone women of reproductive age were less likely among women of primary level of education, aOR = 0.53,95% CI:0.30–0.94;p = 0.029 and those in the poorest wealth index aOR = 0.49,95%CI:0.27–0.88; p = 0.017.

in Sierra Leone at 1.5%, is higher than that reported in the DHS of Kenya (less than 1%) [43], and Uganda (1.3%) [38, 44] but lower than Tanzania (less than 3%) [45].

Studies show that stunting among women of reproductive age is a critical concern, as it reflects long-term exposure to inadequate nutrition, infection, and environmental stress [46]. The consequences of stunting are far-reaching, particularly to girls and women of reproductive

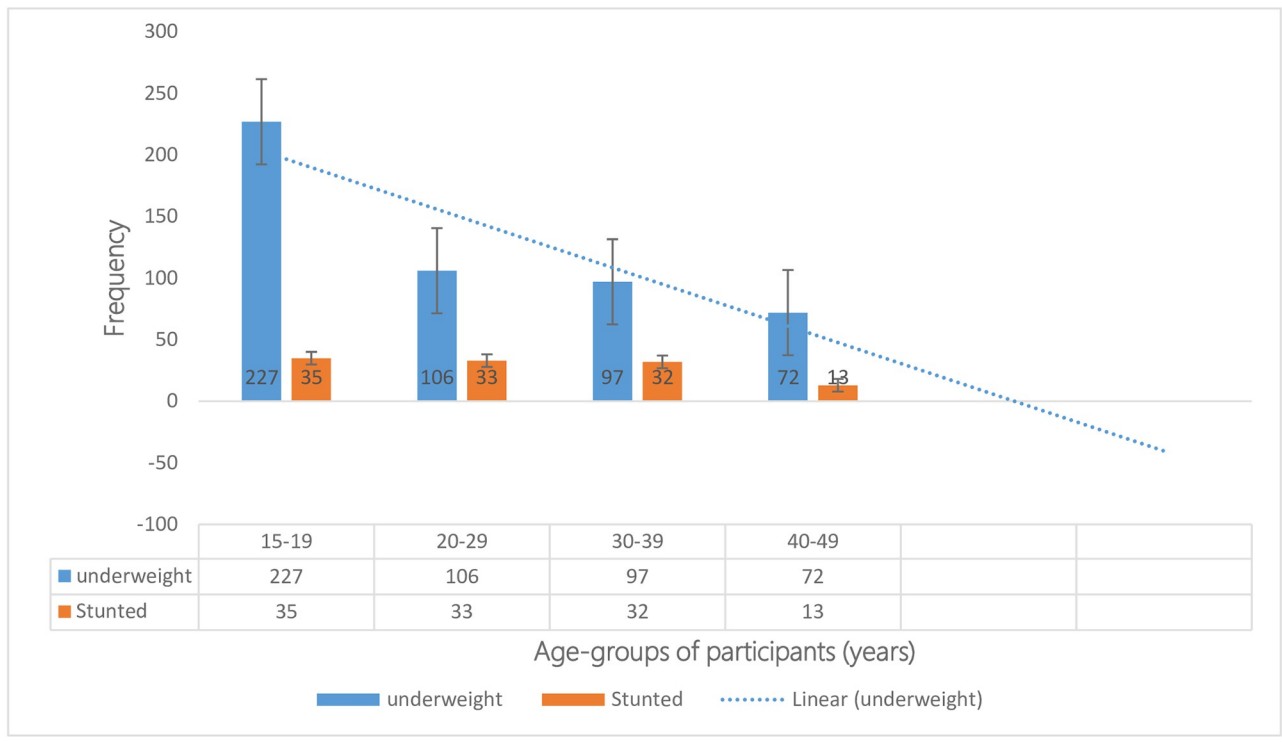

**Fig 3. Frequency of underweight among different age groups of women (15–49 years) in the 2019 SLDHS.** Fig 3 shows the frequency of underweight as it decreased with age group populations, with the majority in the 15-19-year age group 45.2%(227/502), followed by the 20-19-year age group, 21.1%(106/502); 30-39-year age group 19.3%(97/502), and least among the 40-49-year age group 14.3%(72/502). The data source is SLDHS of 2019.

**Table 3. Prevalence and factors associated with underweight among women (15–49 years) in SLDHS-2019.**

| Variables | Under-weight (N = 502) n, % | Normal weight, (N = 4,974), (n, %) | Unadjusted (COR) | 95% CI | P-value | Adjusted (aOR) | 95% CI | P-value |
|---|---|---|---|---|---|---|---|---|
| **Age groups (years)** | | | | | | | | |
| 20–29 | 106(5.6) | 1,773(94.4) | **Reference** | | | **Reference** | | |
| 15–19 | 227(16.0) | 1,192(84.0) | 0.314 | 0.246–0.400 | <.001 | 0.457 | 0.335–0.624 | <.001 |
| 30–39 | 97(7.2) | 1,244(92.8) | 0.767 | 0.577–1.019 | 0.068 | 0.746 | 0.536–1.037 | 0.081 |
| 40–49 | 72(8.6) | 765(91.4) | 0.635 | 0.465–0.867 | 0.004 | 0.663 | 0.453–0.972 | 0.035 |
| **Parity** | | | | | | | | |
| Never gave birth | 225(14.5) | 1,330(85.5) | **Reference** | | | **Reference** | | |
| One to four | 182(6.7) | 2537(93.3) | 2.358 | 1.918–2.899 | <.001 | 1.479 | 1.079–2.029 | 0.015 |
| Five and above | 95(7.9) | 1,107(92.1) | 1.971 | 1.531–2.538 | <.001 | 1.362 | 0.876–2.117 | 0.170 |
| **Residence** | | | | | | | | |
| Rural | 340(9.7) | 3,156(90.3) | **Reference** | | | | | |
| Urban | 162(8.2) | 1,818(91.8) | 1.209 | 0.994–1.470 | 0.057 | | | |
| **Sex of household head** | | | | | | | | |
| Male | 343(8.7) | 3,621(91.3) | **Reference** | | | **Reference** | | |
| Female | 159(10.5) | 1,353(89.5) | 0.806 | 0.661–0.983 | 0.033 | 0.866 | 0.701–1.071 | 0.186 |
| **Household size** | | | | | | | | |
| Six and above | 321(9.7) | 2,998(90.3) | **Reference** | | | | | |
| Less than six | 181(8.4) | 1,976(91.6) | 1.169 | 0.966–1.415 | 0.109 | | | |
| **Work status** | | | | | | | | |
| Not working | 191(11.1) | 1,529(88.9) | **Reference** | | | **Reference** | | |
| Working | 311(8.3) | 3,445(91.7) | 1.384 | 1.144–1.673 | 0.001 | 1.011 | 0.800–1.277 | 0.928 |
| **Marital status** | | | | | | | | |
| Not Married | 270(12.6) | 1,872(87.4) | **Reference** | | | **Reference** | | |
| Married | 232(7.0) | 3,102(93.0) | 1.928 | 1.603–2.319 | <.001 | 1.251 | 0.936–1.672 | 0.130 |
| **Region of residence** | | | | | | | | |
| East | 96(8.1) | 1,082(91.9) | **Reference** | | | **Reference** | | |
| North | 153(10.5) | 1,305(89.5) | 0.757 | 0.579–0.989 | 0.041 | 0.765 | 0.581–1.008 | 0.057 |
| Northwest | 73(9.2) | 724(90.8) | 0.88 | 0.640–1.210 | 0.431 | 0.898 | 0.648–1.243 | 0.515 |
| South | 134(10.3) | 1,173(89.7) | 0.777 | 0.590–1.022 | 0.071 | 0.789 | 0.595–1.045 | 0.098 |
| Western | 46(6.2) | 690(93.8) | 1.331 | 0.925–1.916 | 0.777 | 1.248 | 0.823–1.892 | 0.298 |
| **Level of education** | | | | | | | | |
| Secondary | 185(9.5) | 1,755(90.5) | **Reference** | | | **Reference** | | |
| No formal education | 211(8.1) | 2,399(91.9) | 1.199 | 0.975–1.474 | 0.086 | 0.886 | 0.662–1.186 | 0.417 |
| Primary | 96(12.3) | 686(87.7) | 0.753 | 0.580–0.979 | 0.034 | 0.742 | 0.557–0.989 | 0.042 |

*(Continued)*

**Table 3.** (Continued)

| Variables | Under-weight (N = 502) n, % | Normal weight, (N = 4,974), (n, %) | Unadjusted (COR) | 95% CI | P-value | Adjusted (aOR) | 95% CI | P-value |
|---|---|---|---|---|---|---|---|---|
| Higher | 10(6.9) | 134(93.1) | 1.413 | 0.730–2.733 | 0.305 | 0.677 | 0.338–1.357 | 0.272 |
| **Wealth Indices (WI)** | | | | | | | | |
| Middle | 121(10.3) | 1,050(89.7) | Reference | | | Reference | | |
| Poorest | 104(8.3) | 1,156(91.7) | 1.281 | 0.973–1.666 | 0.078 | 1.236 | 0.929–1.646 | 0.146 |
| Poorer | 120(10.2) | 1,053(89.8) | 1.011 | 0.775–1.320 | 0.935 | 0.935 | 0.711–1.229 | 0.630 |
| Richer | 97(9.1) | 974(90.9) | 1.157 | 0.874–1.533 | 0.309 | 1.150 | 0.850–1.557 | 0.365 |
| Richest | 60(7.5) | 741(92.5) | 1.423 | 1.030–1.967 | 0.032 | 1.158 | 0.782–1.713 | 0.464 |
| **Watching television** | | | | | | | | |
| No | 404(9.5) | 3,851(90.5) | Reference | | | | | |
| Yes | 98(8.0) | 1,123(92.0) | 1.202 | 0.955–1.514 | 0.117 | | | |
| **Listens to radio** | | | | | | | | |
| No | 350(10.4) | 3,007(89.6) | Reference | | | Reference | | |
| Yes | 152(7.2) | 1,967(92.8) | 1.506 | 1.235–1.837 | <.001 | 1.407 | 1.136–1.742 | 0.002 |
| **Reading magazines** | | | | | | | | |
| No | 473(9.1) | 4,698(90.9) | Reference | | | | | |
| Yes | 29(9.5) | 276(90.5) | 0.958 | 0.646–1.421 | 0.832 | | | |
| **Smokes cigarettes** | | | | | | | | |
| No | 484(9.1) | 4,835(90.9) | Reference | | | | | |
| Yes | 18(11.5) | 139(88.5) | 0.773 | 0.469–1.274 | 0.313 | | | |
| **Alcohol use** | | | | | | | | |
| No | 140(6.7) | 2,005(93.3) | Reference | | | | | |
| Yes | 35(7.5) | 429(92.5) | 0.856 | 0.582–1.258 | 0.428 | | | |

aOR: adjusted Odds Ratio; CI: Confidence Interval; COR: Crude Odds Ratio; SLDHS: Sierra Leone Demographic and Health Survey.

In Table 3, the correlates of underweight among Sierra Leone women were likely among women with parity of one-to-four aOR = 1.48,95%CI:1.08–2.03;p = 0.015 and those who listened to radios, aOR = 1.41,95%CI:1.14–1.74;p = 0.002. Being underweight was less likely among age-group of 15–19 years, aOR = 0.46,95%CI:0.34–0.62; p<0.001; age-group of 40–49 years, aOR = 0.66,95%CI:0.45–0.97;p = 0.035, and those with a primary level of education, aOR = 0.74,95%CI:0.56–0.99;p = 0.042

age [47], and the effects are experienced at individual, community, and national levels [48]. It is disturbing to note that an estimated 450 million adult women in developing countries are stunted due to malnutrition during childhood [49]. Thus, addressing stunting among women is critical for improving maternal and child health outcomes.

## Stunting and underweight among women of reproductive age (15–49 years) in Sierra Leone from the 2019 DHS

Women with primary education had 47% less likelihood of being stunted than those with secondary education. Similarly, women in the poorest wealth index had 51% less likelihood of

being stunted compared to those in the middle wealth index. These findings highlight the importance of education and socio-economic status in mitigating the risk of stunting among women. However, no other factors were significantly associated with being stunted in this study population (Table 2).

In contrast, the factors associated with being underweight differed from those of stunting (Tables 2 and 3). A parity of one to four and listening to radios were the significant factors associated with being underweight (Table 3). Women with a parity of one to four were 1.48 times more likely to be underweight than those who had never given birth (Table 3). On the other hand, age groups of 15–19 years and 40–49 years, as well as primary education, were less likely of being underweight. These findings may suggest that different factors contribute to being underweight compared to stunting among women (15–49 years) in Sierra Leone (Table 3) [50].

The underlying reasons for stunting and underweight being less likely among women with primary education in this study population remain unclear, highlighting the need for in-depth exploration through a qualitative research. Conducting qualitative studies would allow deeper understanding of the factors and mechanisms contributing to the observed association between primary education and better nutritional outcomes. By delving into women's lived experiences and socio-cultural context, qualitative research can provide valuable insights to unravel the complex dynamics at play. Thus, additional investigation through qualitative research is warranted to understand why primary education emerges as a shielding factor against being stunted and underweight in this study population.

It is interesting to note that women in the poorest wealth index were less likely to be stunted than women in the middle wealth index (Table 2). This finding contradicts many studies in other African countries where stunting is more prevalent among women in the poorest wealth index [38, 44, 51].

Studies on children five years and below in Sierra Leone from the same SLDHS-2019 show a high prevalence of stunting among this age group [52]. However, our findings that women in the reproductive age group (15–49 years) from the same data source (SLDHS-2019) had no association between stunting and any age group was unique. In contrast, children below five years in Sierra Leone experienced a high prevalence of stunting (31.6% in rural versus 24.0% in urban areas) [52]. This distinctive finding in Sierra Leone necessitates further investigation to explore the underlying factors contributing to these differences among age groups. It is plausible that some low-income households adopted favorable eating habits and practices, such as consuming more locally available foods like "plasas". "Plasas", a mixture of green leaves with palm oil and fish, is affordable and highly nutritious. Understanding the dietary choices and affordability of nutritious foods among low-income households could provide valuable insights into the observed findings.

Furthermore, stunting is a chronic condition that begins during the prenatal period and persists through early childhood and adolescence, with the first two years of life being particularly critical [45, 51]. Previous studies have highlighted the high prevalence of stunting among women of reproductive age in low- and middle-income countries, as stunted children often continue to experience stunting into adulthood [52, 53]. However, it is essential to note that some individuals who were stunted in childhood overcame these challenges by accessing education, obtaining better employment opportunities, increasing their incomes, or marrying into higher socio-economic strata. As a result, some women may have transitioned from lower to higher wealth indices, indicating a potential for social mobility and improvement in their overall nutritional well-being. This socioeconomic progress achieved by some these women may have played an important role in the observed outcome of low socioeconomic status being unlikely of undernutrition (stunting and underweight) in this study population.

In addition, many studies show that improved drinking water was associated with a lower risk of stunting and that improved water was a proxy for less exposure to enteric pathogens [54]. Watanabe and Petri deliberated that environmental enteropathy is a chronic disease caused by continuous exposure to faecally contaminated food and water that does not produce symptoms but contributes to poor physical development [54]. This finding may have been a factor experienced among study populations in other countries but not in Sierra Leone.

These findings on stunting among women in Sierra Leone contrast with another in Uganda, where the population in the Southwestern region (Pygmies and Batwa) were naturally shorter compared to the average Ugandan population [55–57]. More of this could be explained by genetic factors, which play a part at individual level, where it is likely that women of reproductive age in Sierra Leone were generally taller because of their genetic makeup [12]. A contrasting scenario was observed in western Uganda among the pygmies and others who were generally shorter than the average Ugandan population [38, 55, 56]. However, the situation can be determined further by conducting more comprehensive studies on the height profiles of women in Sierra Leone over several decades to determine the changing patterns of women's heights stratified by regions of the country.

In addition, one of the insignificant factors of stunting among women (15–49 years) was the age group of 15–19 years, which is an age group with rapid growth, increased activities, and a higher need for adequate nutrients (Table 2). The need for adequate nutrients and diet among this age group is paramount for their growth and development. Our findings that there were no factors significantly associated with stunting among women in specific age groups and poor household wealth indices were inconsistent with literature from Bangladesh and other countries [57–62].

Genetic predilections and environmental factors mainly determine adult heights [62]. In addition to genetic impacts, incomes, social status, infections, and nutrition have been shown to affect body height in the European population [62]. Also, environmental factors are likely to be more important determinants of height in low-and middle-income countries because environmental stress, including food availability and infections, are higher in those countries compared to high-income countries [58, 59, 60].

Perkins *et al.* explained in their reviews that short adult stature in low-and-middle-income countries is mainly because of cumulative net impacts of malnutrition associated with diseases and environmental conditions, such as socioeconomic status [58].

The factors associated with stunting and underweight among women of reproductive age (15–49 years) in Sierra Leone were different and raised our concerns (Tables 2 and 3). Many factors singly or collectively contribute to underweight and stunting, including eating patterns, food types, their availability, infections, diseases, physical activity levels, and sleep routines [5, 6]. In addition to social determinants of health, genetics and taking certain medications have been shown to play important roles in undernutrition in a population [5, 6, 10, 63].

If compared with overweight and obesity the two are mainly caused by excess food consumption and reduced activities where people gain weight when they eat more calories than they burn through their daily activities [63, 64]. In addition, environmental factors around us matter in developing obesity and overweight, just like stunting and underweight [64]. The world around us influences our ability to maintain a healthy weight and lifestyle [64]. That has been seen in many African communities where people who are obese are considered healthy, living a prosperous and fulfilling life, an issue which is admired by women in many African communities [64].

On the other end of the spectrum, some communities have begun to admire smaller sizes and equate them to successful and healthy lives [64]. In this, several blue-collar workers have begun to reduce their sizes by conducting regular exercises, eating organic foods, fasting,

eating fewer fast foods, less snacking, taking less salts and sugars, living a less sedentary life-style, riding bicycles, or walking to work, sleeping better, avoiding stressful and mental health situations [64].

Perhaps one of the most interesting findings from this study is that the factors associated with underweight and stunting among women in Sierra Leone were different, a factor that should be determined through a comprehensive study, unearthing the underlying reasons. This difference in findings between underweight and stunting in Sierra Leone contrasts with many studies in the African continent [38, 43–45, 47].

Chronic effects of malnutrition in early childhood due to inadequate nutrients and unavail-ability of food are reflected in later life by stunting and other lifelong consequences such as reduced cognitive function, maternal and child health complications, which we did not find in this study population (Table 2).

These factors associated with stunting among Sierra Leone's women ought to be addressed if improvements in maternal and child health indicators are to be achieved soon in this coun-try [64, 65]. Feeding habits, diets, and food availability for young women in Sierra Leone should be prioritized as soon as possible as many young women of reproductive age are affected by stunting and underweight (Tables 2 and 3). In addition, early childhood nutrition programs (for example, school feeding programs) could be a welcome intervention for school-going female children.

It is worth noting that there is limited literature on stunting among women of reproductive age in Sierra Leone, with most studies focusing on underweight. Therefore, findings of this study contribute to this knowledge gap and could be used for setting a proper agenda for the population.

Also, the finding that listening to radios was associated with being underweight could be explained by the effect of group dynamics of a population usually described as an ecological fallacy [66]. Therefore, assessing this association between being underweight and listening to radios among women in Sierra Leone could be unraveled by conducting a qualitative research.

## Strengths and limitations of this study

This study has some strengths. First, the data quality of this study was assured as the SLDHS-2019 used well-trained field personnel, standardized protocols, and validated tools in data col-lection processes. Second, this study utilized a nationally representative sample population of women in the reproductive age of 15–49 years. As a result, the study's findings can be general-izable to the target population in Sierra Leone and other low-to-middle-income countries in the African continent. Third, by using validated tools and calibrated instruments by the SLDHS-2019, the generated estimates are more robust than other studies in Sierra Leone's context. In addition, we used data with a large sample size, which was collected, entered, and cleaned by a team of well-trained and highly experienced scientists, thus limiting mistakes in the dataset used for the analysis. Finally, as we used the concentration index, these findings are more robust in predicting socio-economic inequalities in the study population.

However, this study had limitations that warrant further discussion. First, the SLDHS-2019 was a cross-sectional survey. As a result, we cannot establish a sequential relationship between explanatory and outcome variables. Second, due to the absence of some crucial data, several significant variables, such as food security and dietary diversity could not be included in the final model for the analysis. Third, the SLDHS-2019 did not collect individual incomes and expenditures but household data. It used a wealth index as a proxy indicator for household wealth. Fourth, SLDHS collected data only on 15–49-year-old women of reproductive age in Sierra Leone. With the current changes in adolescents' reproductive actions and behaviors,

there are children less than 15 years old who have gone through the entire cycle of reproduction. As a result, the distribution of undernutrition among women below and beyond this age group (15–49 years) was not factored in the analysis. Finally, most data on correlates of undernutrition were based on self-reported information. They were not verified by using financial records, which risks socially acceptable answers, hence social desirability bias in this result.

## Generalizability of results

Results from this study may be generalized to women of reproductive age (15–49 years) in Sierra Leone and other low-to-middle-income countries.

## Conclusion

The prevalence of underweight and stunting among women of reproductive age (15–49 years) in Sierra Leone was lower compared to regional and world data. This study highlights similarities and differences in this population's prevalence and correlates of undernutrition. Underweight and stunting were less likely in women with primary education, while parity of one-to-four and listening to radios were significantly associated with underweight. Further trend studies using DHS data for 2010, 2014, and 2019 are warranted to understand the dynamics of undernutrition in Sierra Leone.

Furthermore, there is a need to improve the social determinants of health in Sierra Leone among women of reproductive age, including school feeding programs for children and adolescents.

In addition, it is essential to note that this study's findings have important implications for urban-rural maternal and child health in Sierra Leone. The identified correlates of stunting and underweight should be addressed through targeted interventions. Improving feeding habits, ensuring dietary diversity, and addressing food availability for young women in Sierra Leone should be prioritized. Early childhood nutrition programs, such as school feeding programs, could be effective interventions for improving the nutritional status of school-aged female children.

In summary, these findings highlight the importance of education, socioeconomic status, and environmental factors in influencing nutritional outcomes. Addressing the factors associated with stunting and underweight among women is essential for improving maternal and child health indicators in Sierra Leone. Further research is needed to explore the underlying reasons for the observed differences in the factors associated with undernutrition (stunting versus underweight) and develop targeted interventions to alleviate these nutritional challenges.

## Author Contributions

**Conceptualization:** Freddy Wathum Drinkwater Oyat, David Lagoro Kitara.

**Data curation:** Judith Aloyo, David Lagoro Kitara.

**Formal analysis:** David Lagoro Kitara.

**Funding acquisition:** David Lagoro Kitara.

**Investigation:** Eric Nzirakaindi Ikoona, Freddy Wathum Drinkwater Oyat, Emmanuel Olal, Judith Aloyo, David Lagoro Kitara.

**Methodology:** Eric Nzirakaindi Ikoona, Freddy Wathum Drinkwater Oyat, David Lagoro Kitara.

**Project administration:** Nelson Onira Alema, John Bosco Matovu, Emmanuel Olal, Judith Aloyo, David Lagoro Kitara.

**Resources:** Eric Nzirakaindi Ikoona, Amon Njenga, Lucy Namulemo, Ronald Kaluya, Kassim Kamara, Judith Aloyo, David Lagoro Kitara.

**Software:** Eric Nzirakaindi Ikoona, John Bosco Matovu, Lucy Namulemo, Freddy Wathum Drinkwater Oyat, Emmanuel Olal, David Lagoro Kitara.

**Supervision:** Nelson Onira Alema, Eric Nzirakaindi Ikoona, Mame Awa Toure, Amon Njenga, John Bosco Matovu, Lucy Namulemo, Ronald Kaluya, Kassim Kamara, Freddy Wathum Drinkwater Oyat, Judith Aloyo, David Lagoro Kitara.

**Validation:** Nelson Onira Alema, Eric Nzirakaindi Ikoona, Mame Awa Toure, Oliver Eleeza, Amon Njenga, John Bosco Matovu, Lucy Namulemo, Ronald Kaluya, Emmanuel Olal, Judith Aloyo, David Lagoro Kitara.

**Visualization:** Oliver Eleeza, Amon Njenga, John Bosco Matovu, Ronald Kaluya, Kassim Kamara, Freddy Wathum Drinkwater Oyat, David Lagoro Kitara.

**Writing – original draft:** Nelson Onira Alema, Mame Awa Toure, Oliver Eleeza, Amon Njenga, John Bosco Matovu, Lucy Namulemo, Ronald Kaluya, Kassim Kamara, Freddy Wathum Drinkwater Oyat, Emmanuel Olal, Judith Aloyo, David Lagoro Kitara.

**Writing – review & editing:** Nelson Onira Alema, Eric Nzirakaindi Ikoona, Mame Awa Toure, Oliver Eleeza, Amon Njenga, John Bosco Matovu, Lucy Namulemo, Ronald Kaluya, Kassim Kamara, Freddy Wathum Drinkwater Oyat, Emmanuel Olal, Judith Aloyo, David Lagoro Kitara.

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
