## [Editor Report · Decision Letter 0]

20 Sep 2023

PONE-D-23-29763

Prevalence and factors associated with undernutrition among 15–49-year-old women in Sierra Leone: A secondary data analysis of Sierra Leone Demographic Health Survey of 2019.

PLOS ONE

Dear Dr. Kitara,

Thank you for submitting your manuscript to PLOS ONE. After careful consideration, we have decided that your manuscript does not meet our criteria for publication and must therefore be rejected.

The study's selection process, which led to a final sample of 7,514 out of the initial 15,934 women, needs more clarity and transparency. It does not adequately explain why 8,420 women were categorized as having "invalid weight measurements." There is a need for explicit clarification regarding the nature and reasons behind this invalidity, including whether it resulted from misreporting, omissions, or unknown factors.

The study needs to provide more information about its sampling procedure. Readers need a comprehensive description of how the 7,514 women were selected from the larger dataset to ensure transparency and reproducibility. While the authors mention using sample weights to address unequal probability sampling, they do not detail how these weights were calculated or applied. Explaining the methodology used for applying sample weights is essential for transparency.

The study should explicitly state the criteria for "underweight" and "normal weight" to facilitate understanding of the analysis, as it investigates associations between socio-demographic characteristics and women's nutritional status without providing clear definitions for these categories.

In Figure 1 and Figure 2, the source is mentioned as "primary data," but the text refers to the data as coming from the "2019 SLDHS," a secondary data source. This inconsistency creates confusion and should be corrected. Table 1 includes variables related to "work status" and "wealth indices," but the study's title references "socio-demography" without mentioning the economic aspect. The title should accurately reflect the variables included in the table.

 In Table 2, the "Age groups (years) 20-29" as the reference category is not justified.

Providing a rationale for this selection would enhance the understanding of the analysis. In Table 2, it is essential to specify the "reference" category for the variable "working status" to ensure the correct interpretation of the results.

The study should provide a more detailed and comprehensive explanation of the results and their implications in Table 2, as the current interpretation is overly brief and lacks meaningful context, making it challenging for readers to make sense of the findings.

Addressing these critical issues is vital to improving the clarity, transparency, and quality of the study's methodology and reporting. The manuscript has been rejected based on these deficiencies.

I am sorry that we cannot be more positive on this occasion, but hope that you appreciate the reasons for this decision.

Kind regards,

Sathiya Susuman Appunni, Ph D

Academic Editor

PLOS ONE

- - - - -

---

## [Author Response · Author response to Decision Letter 0]

4 Nov 2023

Response to the Academic Editor PLOS ONE.

Title: Prevalence and factors associated with undernutrition among 15–49-year-old women in Sierra Leone: A secondary data analysis of Sierra Leone Demographic Health Survey of 2019.

We want to thank this journal's Academic Editor for the reviews I received after submitting our manuscript on the above title. I have been a reviewer of PLOS ONE for many years, and I have seen the processes and have experience reviewing articles in this journal.

As per our paper presented above, this was a secondary analysis of datasets from the demographic health survey 2019 in Sierra Leone. This data collection was conducted by the Bureau of Statistics of Sierra Leone. Here is A detailed explanation of the final 7,514 respondents out of the 15,934 women.

Methods

Study design: The SLDHS-2019 was conducted as a countrywide representative cross-sectional survey led by the Bureau of Statistics of Sierra Leone (Stats SL) with technical assistance from ICF through DHS programs. This survey was funded by the United States Agency for International Development (USAID)29. 

Study sites: This study was conducted in all four provinces and western areas of Sierra Leone29.

Sampling and study participants: The sampling of the study participants was based on the 2015 population and housing census of the Republic of Sierra Leone30. This census was conducted by Statistics Sierra Leone (Stats SL) and provided the ready-made sampling frame for the SLDHS-201930. 

Sierra Leone is administratively divided into four provinces and western areas (urban and rural), sixteen districts, and 190 chiefdoms30,31,32. Each district is subdivided into chiefdoms/census wards, and each chiefdom/census ward is subdivided into sections30,31,32. In addition, the 2015 population and housing census subdivided each locality into convenient census: the enumeration areas (EAs)30,33. The EAs were the primary sampling units (PSUs) and clusters for the SLDHS-201930-35. The list of EAs from the 2015 census formed the basis for estimating the number of households required and classifying EAs (clusters) into urban/rural for the SLDHS-2019 sampling frame30,31,34,35. 

Furthermore, the SLDHS-2019 employed a two-stage stratified sampling design, and the stratification was achieved by classifying each district into urban and rural areas34,35. So, thirty-one sampling strata were created, and samples were selected independently in each stratum via a two-stage selection process34,35. 

Thus, implicit stratifications were achieved at each lower administrative level by sorting the sampling frame before sample selection according to administrative order and using a probability proportional-to-size selection during the first sampling stage34,35. 

Also, five hundred and seventy-eight (578) EAs were selected using a probability proportional to EA size34,35 in the first stage of the selection process. In addition, the enumeration area size was determined by the number of households residing in it, and a household listing operation was then performed in all selected enumeration areas34,35. The resulting lists of households served as a sampling frame for selecting households in the second stage of the survey34,35.

In the second stage's selection, a fixed number of twenty-four households was chosen in every cluster through an equal probability systematic sampling, resulting in a total sample size of approximately 13,872 households distributed in 578 clusters34,35. The household listing in this stage was conducted using computer tablets, and households were randomly selected through computer programming34,35. 

The survey interviewed only pre-selected households in the clusters, and no replacements or changes of the selected households were allowed in the implementation stage of the survey to prevent selection bias of the study population34,35. Due to the non-proportional allocation of samples to the sixteen districts in Sierra Leone and the possible differences in response rates, sample weights were calculated, added to the data file, and applied so that the results would be representative at national and domain levels34,35. Further, because the SLDHS-2019 sample was a two-stage stratified cluster sampling, sample weights were calculated separately at each sampling stage based on sampling probabilities34,35. After that, the SLDHS-2019 included all women aged 15-49 in the sampled households34,35. Permanent residents in the selected homes and visitors who stayed overnight before the survey were eligible for interviews in the household34,35. The man's questionnaire covered the identification of respondents, background information, reproduction, contraception, marriage and sexual activity, fertility preferences, employment status, gender roles, HIV and AIDS, and other health issues35. The biomarker questionnaire covered the identification of respondents, weights, heights, and hemoglobin measurements for children aged 0–5 years, weights, heights, HIV testing, and hemoglobin measurements for women aged 15–49 years35. The fieldworker questionnaire covered the background information on each field worker35. 

Anthropometric measurements. The weight of respondents was recorded in kilograms (kg) to the nearest decimal point and was measured using an electronic scale (SECA 878)34,35. Participants' heights were measured using a stadiometer in centimeters (cm) to one decimal point34,35. Body Mass Index (BMI) of respondents was calculated in kg/m2 using weights (in kilograms) and heights (meters) of women of reproductive age (15–49 years) and classified according to WHO criteria as underweight (<18.5kg/m2), normal weight (18.5–24.9kg/m2), overweight (25.0–29.9kg/m2), obesity (≥30.0kg/m2 and ≤50.0kg/m2), and overnutrition (≥25.0kg/m2 and ≤50.0kg/m2).

Wealth Index (WI). To calculate each household's wealth, we used the wealth index (WI) as a proxy indicator of household wealth35. This composite index used household key asset ownership variables to calculate each household wealth index from the SLDHS-2019 data35. These variables were the characteristics of the household's dwelling unit, for example, the source of water, type of toilet facilities, type of fuel used for cooking, number of rooms, ownership of livestock, possessions of durable goods, mosquito nets, and primary materials for the floor, roof, and walls of the dwelling place35. The respondent's household wealth index was calculated using computer analysis of household composite factors. It was then categorized into five quintiles: poorest, poorer, middle, richer, and richest wealth indices (Table 1). 

Operational definitions. 

Body Mass Index (BMI): Weight in kilograms divided by heights in meters squared (kg/m2). 

Underweight: BMI <18.5kg/m2

Overweight: BMI ≥25.0kg/m2 and ≤29.9kg/m2 

Obese: BMI ≥30.0kg/m2 and ≤50.0kg/m2 

Overnutrition (Overweight and obese): BMI ≥25.0kg/m2 and ≤50.0kg/m2. 

Enumeration Area (Clusters): An EA is a geographic area consisting of a convenient number of dwelling units that serve as a counting unit for the survey. 

Data Collection: Data collection for this survey was conducted from May 14, 2019, to August 31, 201929. The primary sampling unit (PSU), a cluster, was based on enumeration areas (EAs) obtained from the 2015 EA population census sampling frame29. 

The SLDHS-2019 used five validated questionnaires for the thematic parts of the survey29. The household questionnaire collected data on household environment, assets, and basic demographic information of household members. The woman's questionnaire collected data on women's reproductive health, domestic violence, and nutrition indicators29. The man's questionnaire collected data on men's health, while the biomarker questionnaire collected data on anthropometry and blood tests for mothers and children (0-5 years), and the fieldworker questionnaire collected data on background information of fieldworkers36,37. 

This secondary data analysis included women of reproductive age, 15-49 years, whose anthropometric characteristics were recorded with consent. Trained health technicians were deployed to measure the heights and weights of the participants to ensure the quality of anthropometric measurements29. 

Out of the weighted sample of 15,934 women in the dataset, 7,514 anthropometric measurements were included in the survey design, while 8,420 had invalid weight measurements due to erroneous and ineligible measurements. The weight measurement is vital for calculating the BMI of each participant, which was finally used for assessing the nutritional status of each respondent. 

In some of the participants' results, heights were not well recorded, and we could only obtain completed anthropometric measurements for 7,514 women who were not lactating, non-pregnant, and post-menopausal. In the final analysis, a weighted sample 7,514 was included in our secondary data analysis, as summarized in Table 1. A complete protocol with detailed explanations about data collection processes and sampling is available online29.

Outcome variables: The first outcome variable for this study was stunting. It was coded as "1" for stunted women and "0" for not stunted. Stunting was defined as heights of <145cm ± Standard Deviations (SD) from the median value set by the World Health Organization (WHO). The second outcome variable was underweight, which was defined as BMI<18.5kg/m2 and coded as "1" for underweight women and "0" for normal weight. Normal weight was defined as a BMI of 18.5-24.9kg/m2. 

Independent variables: The independent variables in this study were based on previous studies, the WHO stunting framework, underweight, normal weight, and available information in the SLDHS-2019 database. We included nineteen independent variables in this data analysis. 

Women's characteristics: Parity (categorized as para 0, para one-to-four, and five and above), work status (categorized as working-class versus not working), marital status (categorized as married versus not married/single), levels of education (categorized as no education, primary, secondary, and higher), age groups (categorized as 15-19, 20-29, 30-39, and 40-49 years), woman's stunting status (defined as heights <145cm for stunted and ≥145cm for not stunted women), and woman's BMI classification as normal BMI (18.5-24.9kg/m2) and underweight (<18.5kg/m2).

Household characteristics: These characteristics include regions of Sierra Leone (Northwest, Eastern, Western, Southern, and Northern); household wealth indices (categorized as richest, richer, middle, poorer, and poorest); sex of the head of household (female versus male); household size (less than six versus six and above); residency (urban versus rural); television viewing (yes versus no); reading magazines (yes versus no); listening to radios (yes versus no); smoking cigarettes (yes versus no); and alcohol use (yes versus no).

The study should explicitly state the criteria for "underweight" and "normal weight" to facilitate understanding of the analysis, as it investigates associations between socio-demographic characteristics and women's nutritional status without providing clear definitions for these categories.

In this study, underweight was determined by calculating the body mass index, which is given by the weights (kg) of respondents divided by heights in meters squared (m2) [kg/m2]. The WHO classification of the nutritional status of respondents using BMI, underweight, normal BMI, overweight, and obesity were used in this description.

Underweight is described as BMI<18.5kg/m2, and Normal weight = BMI≥18.5-24.9kg/m2. This explanation has been provided for the method of this revised manuscript.

In Figure 1 and Figure 2, the source is mentioned as "primary data," but the text refers to the data as coming from the "2019 SLDHS," a secondary data source. This inconsistency creates confusion and should be corrected. 

We want to acknowledge it as an error and have revised it to read, "the source of data is SLDHS-2019". We thank you for the advice.

Table 1 includes variables related to "work status" and "wealth indices," but the study's title references "socio-demography" without mentioning the economic aspect. The title should accurately reflect the variables included in the table.

We thank you for the advice. Indeed, we agree that you have revised the title to read "Socio-economic and demographic characteristics."

In Table 2, the "Age groups (years) 20-29" as the reference category is not justified. Providing a rationale for this selection would enhance the understanding of the analysis.

Thank you for your review on this. We have looked at this issue repeatedly and are convinced that using the age group of 20-29 years as a reference category for the analysis was the right decision. This decision is because this age group had a median value, which allowed us to explore the relationship between different age groups with stunting at bi- and multivariable analysis. In the end, there was a significant relationship between the age group of 15-19 years in bivariate analysis but not multivariable regression analysis.

In Table 2, it is essential to specify the "reference" category for the variable "working status" to ensure the correct interpretation of the results.

Thank you for your review. We have noted that we erroneously left out the labeling of the reference category for this variable. We have now included it on the table. The reference category is "not working," and there were no significant relations with stunting.

The study should provide a more detailed and comprehensive explanation of the results and their implications in Table 2, as the current interpretation is overly brief and lacks meaningful context, making it challenging for readers to make sense of the findings.

Thank you for your reviews and advice. We have taken it up entirely and revised the manuscript by including details in the result section.

Addressing these critical issues is vital to improving the study's methodology and reporting's clarity, transparency, and quality. The manuscript has been rejected based on these deficiencies.

We thank you for critically reviewing this manuscript. Because we have provided additional information, we request that you consider re-admitting this revised manuscript for consideration for publication in your journal. We would like to have this article considered by your esteemed journal.

Prof. David Kitara Lagoro

Corresponding author

---

## [Decision Letter · Decision Letter 1]

10 Jan 2024

PONE-D-23-29763R1Prevalence and factors associated with undernutrition among 15–49-year-old women in Sierra Leone: A secondary data analysis of Sierra Leone Demographic Health Survey of 2019.PLOS ONE

Dear Dr. Kitara,

Thank you for submitting your manuscript to PLOS ONE. After careful consideration, we feel that it has merit but does not fully meet PLOS ONE’s publication criteria as it currently stands. Therefore, we invite you to submit a revised version of the manuscript that addresses the points raised during the review process.

**ACADEMIC EDITOR: I am pleased to inform you that two anonymous reviewers have reviewed your manuscript and you are expected to attend to their comments/suggestions as early as you can. Thank you.==============================**

**Please submit your revised manuscript by 15 February, 2024. 11:59 PM. If you will need more time than this to complete your revisions, please reply to this message or contact the journal office at plosone@plos.org. **

**Please include the following items when submitting your revised manuscript:**

**A rebuttal letter that responds to each point raised by the academic editor and reviewer(s). You should upload this letter as a separate file labeled 'Response to Reviewers'.**

**A marked-up copy of your manuscript that highlights changes made to the original version. You should upload this as a separate file labeled 'Revised Manuscript with Track Changes'.**

**An unmarked version of your revised paper without tracked changes. You should upload this as a separate file labeled 'Manuscript'.**

****

**If applicable, we recommend that you deposit your laboratory protocols in protocols.io to enhance the reproducibility of your results. Protocols.io assigns your protocol its own identifier (DOI) so that it can be cited independently in the future. For instructions see: https://journals.plos.org/plosone/s/submission-guidelines#loc-laboratory-protocols. Additionally, PLOS ONE offers an option for publishing peer-reviewed Lab Protocol articles, which describe protocols hosted on protocols.io. Read more information on sharing protocols at https://plos.org/protocols?utm_medium=editorial-email&utm_source=authorletters&utm_campaign=protocols.**

**We look forward to receiving your revised manuscript.**

**Kind regards,**

**Olutosin Ademola Otekunrin**

**Academic Editor**

**PLOS ONE**

**Journal Requirements:**

"No"

6. If possible, please upload a file showing your changes either highlighted or using track changes. This should be uploaded as a Revised Manuscript w/tracked changes, file type. Please follow this link for more information: http://blogs.PLOS.org/everyone/2011/05/10/how-to-submit-your-revised-manuscript/"

7. Your ethics statement should only appear in the Methods section of your manuscript. If your ethics statement is written in any section besides the Methods, please delete it from any other section.

**Additional Editor Comments (if provided):**

****

**Reviewers' comments:**

**Reviewer's Responses to Questions**

**Comments to the Author**

**1. If the authors have adequately addressed your comments raised in a previous round of review and you feel that this manuscript is now acceptable for publication, you may indicate that here to bypass the “Comments to the Author” section, enter your conflict of interest statement in the “Confidential to Editor” section, and submit your "Accept" recommendation.**

**Reviewer #1: (No Response)**

**Reviewer #2: (No Response)**

**2. Is the manuscript technically sound, and do the data support the conclusions?**

**The manuscript must describe a technically sound piece of scientific research with data that supports the conclusions. Experiments must have been conducted rigorously, with appropriate controls, replication, and sample sizes. The conclusions must be drawn appropriately based on the data presented. **

**Reviewer #1: Partly**

**Reviewer #2: Partly**

**3. Has the statistical analysis been performed appropriately and rigorously? **

**Reviewer #1: Yes**

**Reviewer #2: No**

**4. Have the authors made all data underlying the findings in their manuscript fully available?**

**The PLOS Data policy requires authors to make all data underlying the findings described in their manuscript fully available without restriction, with rare exception (please refer to the Data Availability Statement in the manuscript PDF file). The data should be provided as part of the manuscript or its supporting information, or deposited to a public repository. For example, in addition to summary statistics, the data points behind means, medians and variance measures should be available. If there are restrictions on publicly sharing data—e.g. participant privacy or use of data from a third party—those must be specified.**

**Reviewer #1: Yes**

**Reviewer #2: (No Response)**

**5. Is the manuscript presented in an intelligible fashion and written in standard English?**

**PLOS ONE does not copyedit accepted manuscripts, so the language in submitted articles must be clear, correct, and unambiguous. Any typographical or grammatical errors should be corrected at revision, so please note any specific errors here.**

**Reviewer #1: Yes**

**Reviewer #2: Yes**

**6. Review Comments to the Author**

**Please use the space provided to explain your answers to the questions above. You may also include additional comments for the author, including concerns about dual publication, research ethics, or publication ethics. (Please upload your review as an attachment if it exceeds 20,000 characters)**

**Reviewer #1: The data and methods are worthy. The write up is in standard English text but needs to be more concise.**

**All my comments have been added to the PDF and highlighted in yellow. The analysis is detailed but there seems some issue in tables and/ or interpretation of tables. Please recheck the tables and address the issues.**

**Reviewer #2: Comments for the project entitled “Prevalence and factors associated with undernutrition among 15–49-year-old women in Sierra Leone: A secondary data analysis of Sierra Leone Demographic Health Survey of 2019”:**

**1.The whole study based on only four data sets, BMI of reproductive age group females, Wealth proxy index and educational status and parity. Author associated underweight with level of education and wealth index. However, Stunting is a complex condition and dependent upon many factors like, genetic factor, hook warm infection, anemia, vitamin D deficiency, Diet and protein intake, diarrheal diseases, vaccination, age of marriage etc. The study is lacking of data of nutritional status, infection history, psycho-social development status etc. Thus in absence of all such data, interpretation of ‘ lower prevalence of underweight and stunning among reproductive age women in sierra Leone’ may not be correct and needs stronger support for such conclusion.**

**2.The Author mentioned (in result) that 7514 women of reproductive age group participated in the study- which give a wrong impression that it is an active surveillance , whereas it is a secondary data analysis and data of 7514 participants taken from 2019 demographic health survey.**

**3.Out of 15,934 women, selection of 7514 is not properly justified, author mentioned that those responded and within reproductive age group and have data of BMI etc were selected. But did not provided a flow chart for selection. Thus selection bias cannot be ruled out. Rather a detailed flow chart of total participant of the survey ( n=XX), eligible women (mentioning eligibility criteria) ( n=xx), not eligible( n=xxx) and why, total sample available for the study( n=XX) etc. may help to understand the selection process.**

**4.As per the WHO definition of stunning is “height –for-Age of children below 5 years is more than two standard deviations below the WHO’s child growth standards median”. Although author mentioned WHO criteria, but it seems not followed properly as they calculated 113/7514 ( 15-49 years age) as stunning based on weight and height only.**

**5.Control sample is missing. The male of same age group would help to understand the claim of undernutrition and stunning. Also analysis of 0-5 years of age group data may help to support the conclusion.**

**6.The conclusion of the study could not be supported by required data set.**

**7. PLOS authors have the option to publish the peer review history of their article (what does this mean?). If published, this will include your full peer review and any attached files.**

****

**Reviewer #1: No**

**Reviewer #2: No**

****

**While revising your submission, please upload your figure files to the Preflight Analysis and Conversion Engine (PACE) digital diagnostic tool, https://pacev2.apexcovantage.com/. PACE helps ensure that figures meet PLOS requirements. To use PACE, you must first register as a user. Registration is free. Then, login and navigate to the UPLOAD tab, where you will find detailed instructions on how to use the tool. If you encounter any issues or have any questions when using PACE, please email PLOS at figures@plos.org. Please note that Supporting Information files do not need this step.**

---

## [Author Response · Author response to Decision Letter 1]

22 Jan 2024

Response to Reviewers

Prevalence and factors associated with undernutrition among 15–49-year-old women in Sierra Leone: A secondary data analysis of Sierra Leone Demographic Health Survey of 2019.

I wish to thank reviewers for the valuable comments. We are certain that these comments will help us enrich our revised manuscript. We remain grateful to all reviewers.

On the issues raised by reviewers.

1. Thank you, reviewers, for your valuable comments. We wish to state that the two outcome variables (underweight and stunting) were studied against fifteen independent variables (Age, parity, type of residence, sex of the head of household, household size, work status, marital status, regions of Sierra Leone, Level of education, Wealth indices, BMI categories, watching television, listening to radios, reading of magazines, smoking cigarettes, and alcohol use). It is true that we could not exhaust all the variables that affect the dependent variables (Underweight and Stunting) and we agree that these are limitation factors to this study. However, this study was a secondary data analysis which has its own limitations which we acknowledged in this revised manuscript. It is true that Stunting is a complex condition and dependent upon many factors like, genetic factor, hook warm infection, anemia, vitamin D deficiency, diet and protein intake, diarrheal diseases, vaccination, age of marriage etc. However, the overall effects of all these deficiencies are seen in diminished heights and low weight of respondents. In our study, we used the weights and heights to calculate the BMI for each respondent and used the WHO criteria to categorize the BMI into underweight, normal, overweight, and obese respondents. On the other hand, the height was categorized as stunted or not (using <145cm) as the cutoff points for women of reproductive age in Sierra Leone. It is true that the study is lacking data on infection history, psycho-social development status but has information on nutritional data in the BMI results which have been presented in this paper. Furthermore, we, the authors are aware that BMI is a recommended tool by the WHO for assessing the nutritional status in large population studies in spite of its limitations. We, the authors believe that the information provided in this manuscript are a result of a demographic health survey (DHS) in Sierra Leone and it is routinely conducted and published to the wider scientific community and thus, we are confident this manuscript has merits to be published in this journal. Thus, we are certain that even though there are some data that are absent, interpretation of ‘lower prevalence of underweight and stunning among reproductive age women in Sierra Leone’ is appropriate for this study and with the data available.

2. The Author mentioned (in result) that 7514 women of reproductive age group participated in the study- which give a wrong impression that it is an active surveillance, whereas it is a secondary data analysis and data of 7514 participants taken from 2019 demographic health survey. We, the authors, wish to acknowledge and own the mistake we made in that statement. The correct statement should be “this is a secondary data analysis and data of 7514 respondents taken from 2019 demographic health survey was used for this study”. We, the authors sincerely apologize for the errors and have corrected them accordingly in the revised manuscript.

3. Out of 15,934 women, selection of 7514 is not properly justified, author mentioned that those responded and within reproductive age group and have data of BMI etc were selected. But did not provided a flow chart for selection. Thus, selection bias cannot be ruled out. Rather a detailed flow chart of total participant of the survey (n=XX), eligible women (mentioning eligibility criteria) (n=xx), not eligible(n=xxx) and why, total sample available for the study(n=XX) etc. may help to understand the selection process. We, the authors, agree with the comments of the reviewers on this subject. We have now included a flow chart in the revised manuscript to make the information available and clearer. Thank you, reviewers, for your advice.

4. As per the WHO definition of stunning is “height –for-Age of children below 5 years is more than two standard deviations below the WHO’s child growth standards median”. Although the author mentioned WHO criteria, it seems not followed properly as they calculated 113/7514 (15-49 years age) as stunning based on weight and height only. We wish to thank you reviewers for these comments however, we wish to inform you that our study was on women of reproductive age (15-49 years). It is true that we usually use height-for-age for calculating Stunting in children five years and below but in the age group we reported on (women 15-49 years), WHO recommends the use of heights <145cm to define stunting. We are still to learn more from reviewers whether WHO recommends the use of heights-for-age in this age group population we have reported on. Otherwise, we the authors think we have so far done what is recommended by WHO.

5. Control sample is missing. The male of same age group would help to understand the claim of undernutrition and stunning. Also, analysis of 0-5 years of age group data may help to support the conclusion. Thank you, reviewers, for these valuable comments on the control samples. The man’s questionnaire collected information on ½ of households where woman’s survey was conducted. The DHS data tool did not collect the same data for the males as in females and so we were unable to provide the information on the control group. In addition, the unique circumstances of women in the reproductive age in low-resource settings such as Sierra Leone may not completely provide the picture of the nutritional status of men in the same age group. In addition, previous studies showed that the prevalence of underweight and stunting among children five years and below were different from the women (15-49 years). The pattern and prevalence among children below five years are slightly different with those in women of reproductive age (15-49 years). Reference: Sserwanja Q, Kamara K, Mutisya LM, Musaba MW, Ziaei S. Rural and Urban Correlates of Stunting Among Under-Five Children in Sierra Leone: A 2019 Nationwide Cross-Sectional Survey. Nutr Metab Insights. 2021;14:11786388211047056. 

The prevalence of stunting among children five years and below in Sierra Leone from the 2019 DHS was 31.6% (95% CI 29.8-33.2) in rural areas and 24.0% (95% CI 21.6-26.1) in urban areas. The prevalence of stunting among women of reproductive age (15-49 years) from the same DHS of 2019 was 1.5%.

6. The conclusion of the study could not be supported by the required data set. We, the authors, have reviewed the information we provided in the manuscript. We have concluded that what we provided in the revised manuscript reflects the data provided in the result section of the revised manuscript. With due respect, we believe that the conclusion of the study is merited. 

7. Finally, we thank you reviewers for the in-depth reviews which we have found very useful in improving the standard of our revised manuscript. This is most appreciated.

---

## [Decision Letter · Decision Letter 2]

27 Feb 2024

PONE-D-23-29763R2Prevalence and factors associated with undernutrition among 15–49-year-old women in Sierra Leone: A secondary data analysis of Sierra Leone Demographic Health Survey of 2019.PLOS ONE

Dear Dr. Kitara,

Thank you for submitting your manuscript to PLOS ONE. After careful consideration, we feel that it has merit but does not fully meet PLOS ONE’s publication criteria as it currently stands. Therefore, we invite you to submit a revised version of the manuscript that addresses the points raised during the review process.

**ACADEMIC EDITOR: I am pleased to inform you that two anonymous reviewers have reviewed your manuscript. You are expected to address the comments/suggestions of reviewer 1 as soon as possible. Thank you. **

**==============================**

We look forward to receiving your revised manuscript.

Kind regards,

Olutosin Ademola Otekunrin

Academic Editor

PLOS ONE

Journal Requirements:

Reviewers' comments:

Reviewer's Responses to Questions

Comments to the Author

1. If the authors have adequately addressed your comments raised in a previous round of review and you feel that this manuscript is now acceptable for publication, you may indicate that here to bypass the “Comments to the Author” section, enter your conflict of interest statement in the “Confidential to Editor” section, and submit your "Accept" recommendation.

Reviewer #2: All comments have been addressed

Reviewer #3: All comments have been addressed

2. Is the manuscript technically sound, and do the data support the conclusions?

Reviewer #2: Yes

Reviewer #3: Yes

3. Has the statistical analysis been performed appropriately and rigorously? 

Reviewer #2: Yes

Reviewer #3: Yes

4. Have the authors made all data underlying the findings in their manuscript fully available?

Reviewer #2: Yes

Reviewer #3: Yes

5. Is the manuscript presented in an intelligible fashion and written in standard English?

Reviewer #2: Yes

Reviewer #3: Yes

6. Review Comments to the Author

Reviewer #2: Author now addressed all comments reasonably well. Revised version of manuscript is now recommended.

Reviewer #3: REVIEWERS COMMENT-PONE-D-23-29763

Malnutrition, characterized by deficiencies in calories, protein, vitamins, minerals, poor health, and social conditions, poses a significant health challenge for millions of women and adolescent girls worldwide. Adequate nutrition is crucial for women's overall health and has far-reaching implications for the well-being of their children. Children born to malnourished women are at higher risk of cognitive impairments, stunted growth, increased susceptibility to infections, and elevated morbidity and mortality rates throughout their lives. Undernutrition remains a pressing global health issue, encompassing being underweight, wasted, stunted, and with deficiencies in essential minerals and vitamins. Research indicates that women with a body mass index (BMI) below 18.5kg/m2 in developing countries face an escalating mortality risk and heightened vulnerability to illnesses. Despite the significance of understanding maternal nutritional status, limited research has been conducted in Sierra Leone, often focusing solely on malnutrition determinants in young children and adolescents. The present study addresses this research gap by investigating the risk factors of undernutrition among women of reproductive age (15-49 years) in Sierra Leone, utilizing data from the Sierra Leone Demographic Health Survey (SLDHS-2019). Findings of this study hold essential policy implications from a global health perspective and specifically for Sierra Leone, aiding in monitoring progress toward sustainable development goals (SDGs) and regional nutrition strategies. Moreover, the study can guide the allocation of limited resources by the Government and health stakeholders to improve the nutritional and health status of women and infants in Sierra Leone.

Major comment: the study was well researched, detailed and scientific. The methodology, analysis and interpretation were well written and discussed, hence the manuscript is good for publication. However, few corrections are needed.

Minor comments:

The WHO reference criteria for classification of BMI must be provided-Page 6

Page 8-Outcome variables; the WHO reference must be provided

The ethical approval for the protocol (ID number) must be provided on page 9

The first line on page 20 had no reference

References: many of the earlier references on pages 24 and 25 are obsolete.

7. PLOS authors have the option to publish the peer review history of their article (what does this mean?). If published, this will include your full peer review and any attached files.

Do you want your identity to be public for this peer review? For information about this choice, including consent withdrawal, please see our Privacy Policy.

Reviewer #2: Yes: Dr. Madhuchhanda Das

Reviewer #3: No

---

## [Author Response · Author response to Decision Letter 2]

25 Aug 2024

Point by point response to reviewers.

Prevalence and factors associated with undernutrition among 15–49-year-old women in Sierra Leone: A secondary data analysis of Sierra Leone Demographic Health Survey of 2019.

On behalf of the authors, I wish to thank you reviewers for the comprehensive review of our manuscript. We are certain it has helped us to improve the quality of our manuscript. Your reviews are well appreciated.

The point-by-point response.

1. The WHO reference criteria for classification of BMI must be provided-Page 6. We agree with your suggestion and on page 6 we have included this. The Body Mass Index (BMI) of respondents was calculated in kg/m2 using weights (in kilograms) and heights (meters) of women of reproductive age (15–49 years) and classified according to WHO criteria as underweight (<18.5kg/m2), normal weight (18.5–24.9kg/m2), overweight (25.0–29.9kg/m2), obesity (≥30.0kg/m2 and ≤50.0kg/m2), and overnutrition (≥25.0kg/m2 and ≤50.0kg/m2) [36]. [36] World Health organization (WHO). Malnutrition. 2023. https://www.who.int/news-room/fact-sheets/detail/malnutrition.

2. Page 8-Outcome variables; the WHO reference must be provided. We agree with your suggestion and thank you very much. Here it is. The first outcome variable for this study was stunting among women (15-49 years). It was defined as heights of <145cm ± Standard Deviations (SD) from the median value set by the World Health Organization (WHO) [37,38]. [37] WHO. Global nutrition targets 2015: stunting policy brief (WHO/NMH/NHD/14.3). World Health Organization: Geneva. 2014a. [38] Sserwanja Q, Mukunya D, Habumugisha T, Mutisya LM, Tuke R, Olal E. Factors associated with undernutrition among 20-to 49-year-old women in Uganda: a secondary analysis of the Uganda demographic health survey 2016. BMC Public Health. 2020;20:1644. 

3. The ethical approval for the protocol (ID number) must be provided on page 9. Yes, we agree that we should provide the ethical approval number for this survey. We acknowledge this as an important requirement for this publication. However, we are requesting you the reviewer that when we were given the authorization to use the Sierra Leone DHS data, it was without the IRB approval number which was conducted by the Sierra Leone Ethics and Scientific Review Committee and the IRB of ICF. For more information, I wish to refer you to these documents and the letter of authorization to use this DHS data of 2019. https://dhsprogram.com/pubs/pdf/FR365/FR365.pdf and Quraish Sserwanja, Kassim Kamara, Linet M Mutisya, Milton W Musaba and Shirin Ziaei. Rural and Urban Correlates of Stunting Among Under Five Children in Sierra Leone: A 2019 Nationwide Cross-Sectional. Nutrition and Metabolic Insights. 2021;4:1–10. We contacted the ICF program manager for the IRB and have not yet sent it to us. We therefore request you to allow this manuscript to proceed as this was an international survey which has been regularly conducted across countries and there has been a number of publications that have been made from secondary data analysis of DHS data.

4. The first line on page 20 had no reference. We agree to your review comment on page 20. We have reviewed it and included the citation and reference. The reference is [54]. [54] Watanabe K & Petri WA Jr. Environmental enteropathy: elusive but significant subclinical abnormalities in developing countries. EBioMedicine. 2016;10:25–32.

5. References: many of the earlier references on pages 24 and 25 are obsolete. We agree to your review comments and have accordingly revised references in pages 24 and 25 and are here for you to review our work. Overall, we are grateful for your review comments and thank you very much.

---

## [Decision Letter · Decision Letter 3]

25 Sep 2024

Prevalence and factors associated with undernutrition among 15–49-year-old women in Sierra Leone: A secondary data analysis of Sierra Leone Demographic Health Survey of 2019.

PONE-D-23-29763R3

Dear Dr. Kitara,

We’re pleased to inform you that your manuscript has been judged scientifically suitable for publication and will be formally accepted for publication once it meets all outstanding technical requirements.

Kind regards,

Olutosin Ademola Otekunrin

Academic Editor

PLOS ONE

Additional Editor Comments (optional):

Reviewers' comments:

Reviewer's Responses to Questions

**Comments to the Author**

1. If the authors have adequately addressed your comments raised in a previous round of review and you feel that this manuscript is now acceptable for publication, you may indicate that here to bypass the “Comments to the Author” section, enter your conflict of interest statement in the “Confidential to Editor” section, and submit your "Accept" recommendation.

Reviewer #3: All comments have been addressed

2. Is the manuscript technically sound, and do the data support the conclusions?

Reviewer #3: Yes

3. Has the statistical analysis been performed appropriately and rigorously? 

Reviewer #3: Yes

4. Have the authors made all data underlying the findings in their manuscript fully available?

Reviewer #3: Yes

5. Is the manuscript presented in an intelligible fashion and written in standard English?

Reviewer #3: Yes

6. Review Comments to the Author

Reviewer #3: we appreciate the authors for painstakingly addressing all issues raised during the course of reviewing this manuscript. indeed, the output is applaudable. i wish to recommend that the manuscript proceed to the publication phase.

7. PLOS authors have the option to publish the peer review history of their article (what does this mean?). If published, this will include your full peer review and any attached files.

Reviewer #3: No

---

## [Editor Report · Acceptance letter]

9 Oct 2024

PONE-D-23-29763R3 

PLOS ONE

Dear Dr. Kitara, 

I'm pleased to inform you that your manuscript has been deemed suitable for publication in PLOS ONE. Congratulations! Your manuscript is now being handed over to our production team.

Kind regards, 

on behalf of

Dr. Olutosin Ademola Otekunrin 

Academic Editor

PLOS ONE